# Factors Influencing Business Analytics Solutions and Views on Business Problems

**Martin Potančok** [1,*] **, Jan Pour** [1] **and Wui Ip** [2]

1 Department of Information Technologies, Faculty of Informatics and Statistics, Prague University of Economics and Business, nám. W. Churchilla 1938/4, 130 67 Prague 3, Czech Republic; jan.pour@vse.cz

2 Department of Pediatrics, School of Medicine, Stanford University, 300 Pasteur Drive, MC: 5776, Stanford, CA 94305-5776, USA; wui@stanford.edu

\* Correspondence: martin.potancok@vse.cz

**Abstract:** The main aim of this paper is to identify and specify factors that influence business analytics. A factor in this context refers to any significant characteristic that defines the environment in which business analytics and business in general are conducted. Factors and their understanding are essential for the quality of final business analytics solutions, given their complexity and interconnectedness. Factors play an extremely important role in analytic thinking and business analysts' skills and knowledge. These factors determine effective approaches and procedures for business analytics, and, in some cases, they also aid in the decision to delay a business analytics solution given a situation. This paper has used the case study method, a qualitative research method, due to the need to carry out investigation within the actual business (company) environment, in order to be able to fully understand and verify factors affecting analytics from the viewpoint of all stakeholders. This study provides a set of 15 factors from business, company, and market environments, including their importance in business analytics.

**Keywords:** business intelligence; data; factor analysis; information system development; management practices; organizational culture

## 1. Introduction

The focus of this paper is the business analytics environment; especially factors that can influence such environments in the broader context of each analytics solution and thus prevent unnecessary failures or disappointments. Activities related to business analytics require an analytical environment, and analytical thinking by all stakeholders [1,2]. The basis of such thinking is an analytical view of business in general and an analytical perception of the company. It is necessary to emphasize that in this context, it is not just a question of solving the tasks of business analytics itself [3], but of a broader view of the company's development, sources, limits, threats etc. [4]. The basic principles and areas of analytical thinking are intuition, creativity, common sense, and analytical knowledge of business content [5]. Business problems are always specific and related to the company environment, e.g., a lack of interest from the management, a lack of attention to the quality of the input data, or a lack of time and motivation for managers to devote themselves to more demanding and sophisticated analytical solutions.

There is an increasing number of analytical services and their capabilities, and a growing market of providers (in 2019 Gartner projected a growth of 12.9% in the total value of $267 billion [6]). Although business analytics is currently in the Plateau of Productivity stage on the Gartner's Hype Cycle [7], there are emerging subfields (e.g., immersive analytics [8], analytics catalogue [9], or continuous intelligence [10]), which create new opportunities for business analytics, as well as needs to refine existing practices. No matter how projects, including business analytics ones, are solved, our experience and the findings

of other authors [11,12] have shown that a good understanding and knowledge of the factors that influence the solution have a positive effect on the final result.

We define a factor as any essential characteristic that defines or evaluates the environment in which business analytics is implemented. This definition is based on the theory of information strategy [13] and extends the analytics theory defined as: "Analytics is the process of developing actionable insights through problem definition and the application of statistical models and analysis against existing and/or simulated future data" [14].

The structure of this paper corresponds to what was stated above. Firstly, we define research questions using the rationale underlying this study and the relevant data from companies. Secondly, the research design, which uses the case study method, is described. Thirdly, the fundamental structure of the relevant factors that impact business analytics solutions in various management fields is presented together with the individual factors and their importance in business analytics.

## 2. Literature Review

The business analytics environment and its components, applications, sources, and approaches have been studied and described by several authors. The most comprehensive studies have been provided by Slánský [15], Novotný et al. [16] and Evans [17]. They identified the value of business analytics and its elements for a given company. All these authors have agreed that the main goal of business analytics is decision support. This fits into the following definition: "delivering the right decision support to the right people and digital processes at the right time" [3].

On a business level, there is a large number of such factors in every specific company environment and its development [18,19]. Examples include identified customer expectations, return on capital employed, competition, and sustainability. In their case studies, Gaardboe and Svarre [20] analyzed the critical success factors from the following perspectives of task, people, structure, and technology from a Business Intelligence (BI) perspective. Hung et al. described how technical, organizational, and environment-related characteristics affect user satisfaction and the overall system effectiveness [21]. A positive impact of BI infrastructure, leadership, and financial commitment on a company's success has been demonstrated by the study of Gonzales et al. [22]. T. Ramakrishnan et al. have shown a positive relationship between institutional isomorphism and implementing BI for the purpose of achieving consistency and initiating a comprehensive BI data collection strategy; implementing BI for the purpose of achieving organizational transformation; and initiating a comprehensive BI data collection strategy [23]. In relation to the purpose, and contribution to the decision-making culture, Sparks and McCann [24] showed the importance of information-content quality and information-access quality.

Business Intelligence is a specific part of business analytics. As it is a sub-part of it, authors generally do not cover all the factors that influence business analytics solutions and views on business problems [3]. An example might be the research on business analytics adoption within specific environments (companies, countries etc.) [25,26]. It is therefore essential to study business analytics as such, wherein the research is currently missing and only partial outputs can be encountered, such as the organizational context, development process, and implementation [27]. The main aim of this paper is to identify factors that influence business analytics. When studying these factors, specific features of the business environment have been taken into account. This article reflects the current situation in business analytics research and the impacts of the continually expanding possibilities as mentioned above. The following research questions were posed:

1.    What factors influence business analytics?
2.    How strong is the impact of each of the factors on business analytics?

## 3. Research Methodology

This paper has used the case study method, the qualitative research method, and qualitative content analysis [28], due to the need to carry out investigation within the actual

business (company) environment, in order to be able to fully understand and verify the defined basic set of factors affecting analytics from the viewpoint of all stakeholders.

The research design and the case study procedures for this paper are based on the research design used by Lacity et al. [29] and Potančok and Voříšek [30] This approach is based on initial sampling (for this paper, we determined an initial set of factors influencing business analytics from the literature) and personal interviews with stakeholders, with the aim to provide specific evidence for identifying key factors. The need to study the factors that influence analytics within the context of a real environment was the most significant reason for choosing this research design.

All the most important topics (in a schematic form) of business analytics (BA) solutions [15] for the sectors of banking, government, healthcare, industry, retail, and transport are shown in Table 1. These are topics (the basic building blocks, derived from the literature mentioned above) that must be addressed by business analytics. They form the basic model of business analytics. Understanding business analytics and the individual topics will serve for a more detailed analysis of the factors during interviews and subsequent analyzes.

**Table 1.** Business Analytics Topics and Structures.

| Topic | Structure |
|---|---|
| Basic Concepts | Business Intelligence, Self Service Business Intelligence, Competitive Intelligence |
| Components | Data Warehouse, Data Mart, Extract Transform Load (ETL/ELT), OLAP Database, Data Staging Area, Tabular model, Data Lake, Sandbox, Real-Time Data Warehouse, Mobile BI, In-Memory Analytics |
| Reporting and Visualization | Reporting, Dashboards, Data Visualization |
| Applications | Analytic Applications, Planning Applications, Collaborative Decision-Making, Business Activity Monitoring |
| Data Science and advanced analytics | Data Science, Data Mining, Text Mining, Predictive Analytics |
| Data sources | Master Data Management, Data Quality, Data Profiling, Big Data |
| Corporate Performance Management | Corporate Performance Management, IT Performance Management, eGovernment Performance Management, Sales Performance Management |
| Approaches | Waterfall Concept, Agile Concept |

Source: Prepared by the authors.

Business analytics is applied in various fields of company management based on various functionalities and various effects, focused on various stakeholders and thus respecting various impact factors. Various fields of company management include the individual topics of business management. The basic relationships of management fields—factors—business analytics are documented in Figure 1.

The stakeholders of this study include analysts from companies based in the Czech Republic. In the course of the case study, interviews were conducted at the turn of 2019 and 2020. During the case study, 25 interviews with randomly selected employees, users of business analytics, senior managers, and team members of business analytics from the sectors of banking, government, healthcare, industry, retail, and transport were conducted. The length of the interviews ranged from 30 min (the indirect participants—employees and users of business analytics) to 1.5 h (the direct participants—the senior managers and team members of business analytics). The beginning of each interview was unstructured, in order to get as much information and as many opinions as possible, followed by a semi-structured part with questions drawing on the basic factors influencing business analytics. Other sources of information included several types of information and data strategies, internal notes, meeting minutes, annual reports, and the companies' organizational structure.

This application is fully consistent with the definition of a case study as a qualitative research method [31,32].

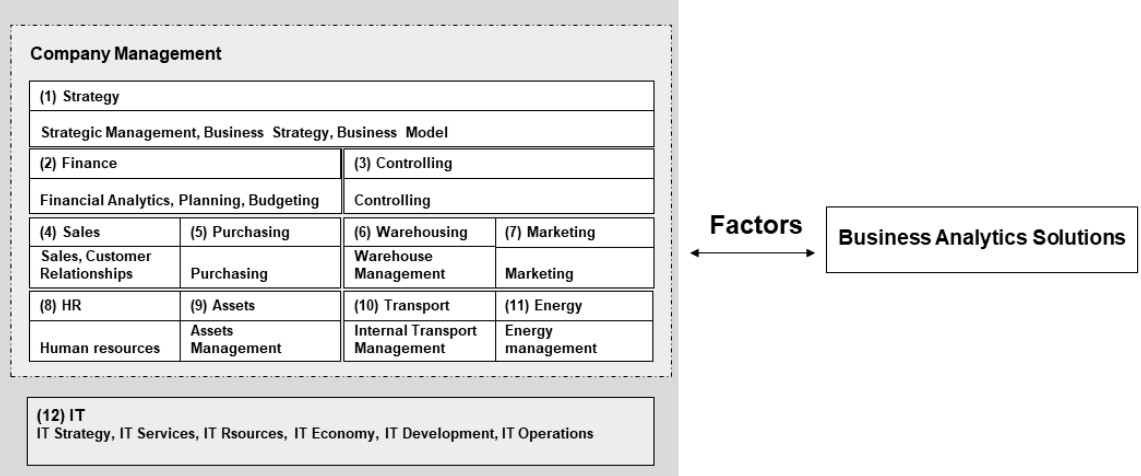

**Figure 1.** Management Fields—Factors—Business Analytics.

## 4. Results and Discussion

Based on the initial sampling and personal interviews with stakeholders, we identified 15 key factors influencing analytics. This set contains the following factors (sorted alphabetically): Applications, Application Resources; Applied Approaches and Methods; Availability, Quality, and Price Range; Business Partners; Cloud Computing; the Company and its Characteristics; Company Culture; Current State of Business Analytics; Data Sources; Human Resources; IT infrastructure; Legislation; Products; Services; and Sourcing.

Subsequently, we must define the basic structure of the relevant factors impacting business analytics solutions in the various management fields presented in Figure 1. A summary of the relevant factors in the context of business content and business analytics solutions is shown in Figure 2. The factors are categorized into 3 main groups: Business Environment, Company Environment, and Market Environment. The following paragraphs describe all the factors in more details.

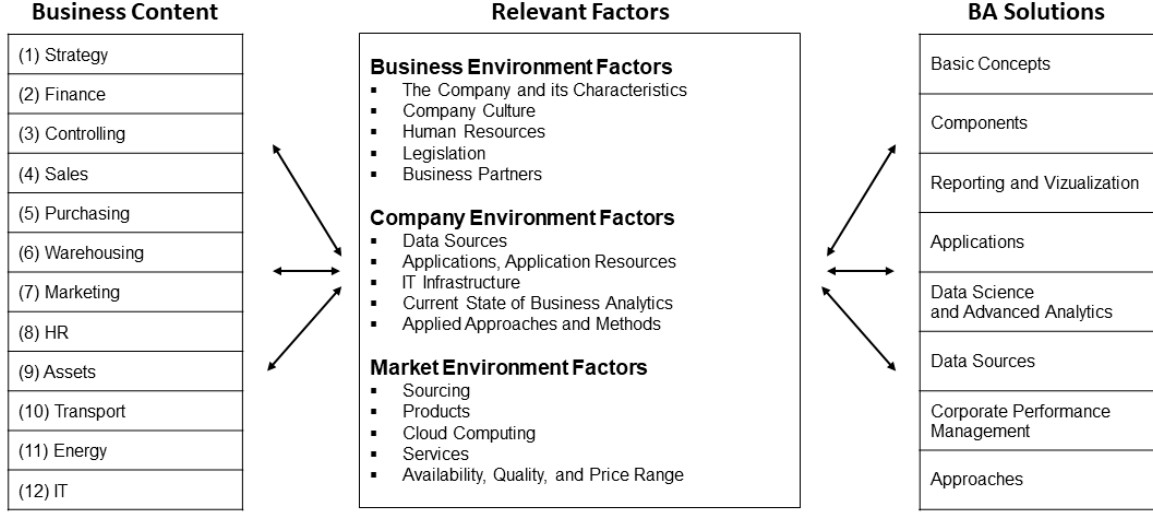

**Figure 2.** Factors impacting business analytics use in company management.

### 4.1. Business Environment Factors

The purpose of monitoring and evaluating the factors of the business environment (i.e., Company, Company Culture, Human Resources, Legislation, Business Partners) is to understand the real needs of a company for business analytics, depending on its size, industry orientation, and market activities, including the needs of individual managers.

### 4.1.1. The Company and Its Characteristics

Significant characteristics of the company itself (i.e., size, industry, ownership, competitive environment) affect the success of business analytics. The size of the company is usually determined by the number of employees and the annual turnover. Small companies (with 1 to 100 employees and an annual turnover not exceeding CZK 30 million) have simpler management, fewer users, but also limited financial resources, which means the owners are generally more careful about investing in business analytics. Medium-sized companies (with 101 to 500 employees and a total turnover of 31 to 100 million CZK) are characterized by a significantly higher increase in business analytics applications, which is due to increased knowledge of employees and more availability of technology. For large companies (with more than 500 employees and a turnover of more than CZK 100 million), it is typical to invest large sums of money in BA and allocate considerable human resources to it [33]. These companies manage huge volumes of data in data warehouses, requiring significant demands on their management with additional complexity of managing the entire IT infrastructure. For large companies, given their significant investment in financial and human capital, the environment is better suited for applying advanced analytics, including data mining, predictive analytics, Competitive Intelligence, and other advanced methods in data science.

The branch of activity of the company is based on the NACE—Classification of Economic Activities. This is especially important for the content of corporate analytics applications, as it will be demanding in its functionality [34]. Firms in various sectors of the economy, with their complexity of management, pressure on efficiency, range of diverse data sources and existing IT infrastructure, create preconditions for the development of business analytics with varying demands on functionality and the technology used.

Business ownership is one of the factors that expresses the forms and complexity of ownership relationships. There is a difference between Czech-owned companies and those that are a branch of a multinational company, or exclusively a foreign company operating in the Czech environment and on the Czech markets. In smaller (local) companies, owners are, at the same time, managers; therefore, they are often involved in business analytics. For large (international) companies, with a high number of owners, this issue is more complex.

The competitive environment of companies is described in several publications and models (e.g., Porter's model of competition). In this context, it is a factor that must be considered when planning and dealing with IT development. IT typically tends to become significantly strengthened, while also undergoing major changes in its structures, the strength of influence, and the subjects that enter it in various forms at the same time.

### 4.1.2. Company Culture

The system of values that the company professes affects the management style and the level of detail in which issues are addressed in the company (e.g., how detailed the individual processes are) [35]. Company culture, the nature of management, and development orientation are a prerequisite or, on the contrary, a limitation for the application of business analytics.

The level of management is a complex factor influencing not only the company's success on the market, but also the qualified and effective use of business analytics. The evaluation of the level of management includes the level of human resources for management, organization settings, and communication maturity.

The essence of the Innovation Management factor lies in defining the significance of innovations for the existence of the company, determining the method of their management and analysis of related problems. According to Drucker [29,36], "companies adapt to the changing business environment by improving their products, services, processes, production technologies or by changing the entire business model."

Management's views on analytics are a factor that still emphasizes the support and perception of responsibility for the success of the solution.

Standardization and formalization in the company is a standard method for the development of the company, especially in the processes of company management, in production, sales activities, and IT. Business analysts generally should take standardization and formalization into consideration [37].

### 4.1.3. Human Resources

The functions and roles of employees should be relatively well defined. Roles are usually defined by their functional content, qualification requirements, and knowledge and competencies. Business analytics is an area that is evolving and changing rapidly, and these changes should be included in the relevant role (job) description.

Business analytics projects are a type of project wherein the users' ownership of their solutions is significantly higher than in standard transaction application projects [13]. This in itself carries a corresponding need for preparing users and knowing the working methodologies, products, and their functionality, nowadays very often discussed as data literacy [38,39].

### 4.1.4. Legislation

Legislation includes regulatory requirements for reporting and analysis of legislative impact on business [40]. This factor represents a summary of the impacts of laws and standards, and all legislative restrictions that must be observed at the application level. The impact of legislation and its constant changes on business analytics is less than that of operations of a transactional nature, but it is still particularly noticeable in data work.

### 4.1.5. Business Partners

The Business Partners factor includes customer and supplier claims on analytical information. The factors of the environment, i.e., the whole complex of Business Partners, are reflected primarily in applications providing external relations and communication (Customer Relationship Management (CRM), Supplier Relationship Management (SRM), Supply Chain Management (SCM)) [33].

### *4.2. Company Environment Factors*

The purpose of monitoring and evaluating factors of the company environment is to analyze the available resources for business analytics. In the main, it is useful to include the following factors in this group: Data sources (range of data sources, availability, quality, use of external data sources); Applications, application sources (complexity, maturity, modularity of products and processes in the company, degree of integration, expected application development in company); IT infrastructure (its quality for business analytics); Current effectiveness of internal analytics (economic comparison of providers and internal options); and Applied approaches and methods of analytics management (at all levels, setting priorities, attitude to knowledge transfer, systematic thinking).

### 4.2.1. Data Sources

When assessing the possibilities and preconditions for the development of business analytics, relevant aspects include the scope of data sources, the possibility of direct access to databases and data extraction (in CSV formats or XML files), combinations of internal and external sources, and evaluation of the availability and quality of resources, including their documentation [15].

### 4.2.2. Applications, Application Resources

This factor includes a number of elements related primarily to information and thus also to the company's global strategy, which includes the company's goals and their priorities, the requirements of the process owners, and the relationship to the company's environment [41,42].

### 4.2.3. IT Infrastructure

The IT infrastructure forms the basis for analytical services and environments for data storage, processing, and access. The main focus of most companies is not business analytics itself, so there is not much emphasis on experience with IT infrastructure. According to the results of a case study, 80% of operational IT requirements on the part of users can be handled by a medium experienced IT specialist [30]. The factor is therefore most pronounced in design, development, and implementation of business analytics solution or services.

Information about the IT infrastructure can be obtained from the evaluation of individual IT department employees or configuration management databases (CMDB) [43]. The obtained results of the factor must be compared with the business and functional requirements. Based on this evaluation, the priority can be either expanding one's own infrastructure or using the services of external partners (if the possibilities, maturity, experience, knowledge, and skills do not meet the requirements).

### 4.2.4. Current State of Business Analytics

This factor relates to the degree of integration of analytics with the reality of the business [44]. Its consideration is essential for the development of and approach to providing business analytics.

The identification of the degree of integration can be performed on the basis of the model of IT enterprise integration (IT integration with business and IT integration). During this identification, the level of integration (technological, methodological, value, internal business processes, with the company's environment, vision, values, processes, and IT services) that the company wants to achieve is determined. The influence of the degree of integration can be observed according to the chosen integration model [44].

### 4.2.5. Applied Approaches and Methods

Applied approaches and methods influence the concept and approach of the leading analyst at all levels. The degree of standardization and formalization is essential for them. In the case of internal ones, internal procedures are created in the company, while in the generally accepted ones, nationally or internationally valid approaches and methods are used. It is necessary to consider internationally recognized approaches and compare implementations in the company with similar implementations on an international scale.

In cases where the company uses internationally recognized approaches, the transfer of services, processes and resources to an external partner [45] is easier. Standardization and the degree of the formalization [37] of the approaches and methods used facilitate and support outsourcing.

### *4.3. Market Environment Factors*

The purpose of monitoring and evaluating the factors of the Market Environment is to continuously analyze the available external resources on the IT market, especially the availability and quality of services and products. In the main, it is useful to include the following factors in this group: Sourcing in business analytics (availability of quality and experienced staff); Offer of products for business analytics, its integration into other products (Enterprise Resource Planning (ERP), CRM); Cloud Computing (the range of services specific to business analytics in the Cloud); Supply and availability of analytical services on the market, availability of a provider, provider's capabilities, and support of technologies; Quality and cost of analytical services.

### 4.3.1. Sourcing

Sourcing (sourcing strategy) [46] means the quantitative and qualitative availability of people on the market who fully understand the field of business analytics. This translates into a comprehensive range of views on business analytics, from strategic business, through economic and technological aspects, to user analytics. A good overview of business

analytics requires a combination of experience and knowledge from all spheres. In most cases, the overall knowledge portfolio accumulates within teams, not with individuals, because the topic is so broad that relying only on individuals can be very risky.

### 4.3.2. Products

In using the term "product range", we mean a technological solution in which we want to ultimately implement our business analytics. Each of the products has its own focus and functionality and can help in its own way; it is always necessary to consider them in relation to the business's own needs, understand them, and then look for a solution at the technical level.

In general, two sources can be recommended for tracking business analytics products in today's market. The first is the annual Gartner report on BI tools, an analysis of who is at the forefront and why [9]. The Gartner report also describes expected trends. The second key source of information is the BARC Research annual study [47], which evaluates the BI tools from the perspective of user needs.

BI products are application software used for the collection and subsequent processing of various types of data. As a result, these products can combine a wide range of resulting data components for analysis, including ad hoc queries, report generators, BI for mobile devices, and real-time business analytics. It includes visualization software with reports and dashboards for key indicators that we want to evaluate in business analytics.

### 4.3.3. Cloud Computing

A special form of a product solution is the implementation of business analytics in the Cloud environment [48]. Cloud Computing in business analytics provides not only a fast and secure network service, but also software and hardware that is customizable according to a company's need, including server, network, storage, various software applications and services, etc.

Cloud Computing provides computing, storage, services, and applications over the Internet (Cloud)—servers, storage, databases, networks, software, analytics, intelligence, etc., [49]. Given Cloud Computing typically charges end users based on the level of usage, it helps reduce traffic costs and allows companies to operate infrastructure more efficiently, and scale according to changing business needs.

### 4.3.4. Services

The topic does not end with the offer of business analytics products on the supplier market. Services are inextricably linked to products. Services may be product specific or general to any analytics topic.

The use of supplier services (services from the provider [50]) is in many cases a more advantageous investment than trying to cover a topic by buying a tool and employing/educating one's own people in that field. It always depends on the business and its scope. In many cases, it makes sense to combine internal and external resources, at least for the first pilot phases, before the company learns to work independently through the scheme.

### 4.3.5. Availability, Quality and Price Range

Taking into account all the previous factors of the Market Environment, the combination of quality, price level, and availability, both for services and for products or human resources, will play a role in the context of simpler requirements and possibilities.

The availability of a product overview is crucial for us, especially when we need to ensure the operation of the product, as well as in situations where we will have requirements for specific technological solutions that cannot do without a unified consultation and solution.

From the point of view of services, availability is essential for a company if we go into the topic of business analytics accompanied by external suppliers and, at least at the

beginning, we need help from them in the form of proof of concept and consultations. Services are also essential for us when we need to cover the technical support of the system and it pays to outsource this task to suppliers.

The findings of the case study are summarized in Table 2. The factors are listed in alphabetical order in each group (internal, internal/external, and external). This classification into groups is from the perspective of the company; internal factors or organizational factors arise within the organization and can be directly influenced. External factors come from the external environment and can be influenced only indirectly. This information will help companies build on the factors. More details are included below in Section 5.

**Table 2.** Factors influencing business analytics.

| Subject Area/Environment | Factor | Classification |
| --- | --- | --- |
| Business | Company | Internal/External |
| Business | Company culture | Internal |
| Business | Human resources | Internal/External |
| Business | Legislation | External |
| Business | Business Partners | Internal/External |
| Company | Data sources | Internal |
| Company | Applications, application resources | Internal |
| Company | IT infrastructure | Internal |
| Company | Current state of business analytics | Internal |
| Company | Applied approaches and methods | Internal |
| Market | Sourcing | Internal/External |
| Market | Products | External |
| Market | Cloud Computing | External |
| Market | Services | External |
| Market | Availability, quality and price range | External |

Source: Prepared by the authors.

## 5. Managerial Implications

During the case study, it was possible to find different views on the individual factors and their importance in business analytics. The importance of a factor can be understood as a reflection of the extent to which the business analytics is influenced by it. The above-mentioned factors influence the final strategy in an uneven manner, and therefore, it is necessary to determine the extent of their impact and importance. The method of pairwise comparison with matching indifference of factors without elimination [51,52] was used to determine the factors' impact. In accordance with the pairwise comparison method, each of the elements (factors) was compared with all the others; 1 point was assigned to the winning element, 0 to the losing one, and 0.5 was assigned in case of a tie. The results were then divided into 3 categories where the strength of impact values is as follows:

- 3: The factor influences the use of BA very strongly; it must be analyzed in more detail. For instance, the "company culture" has a fundamental impact on the BA solutions in business strategy or sales management.
- 2: The factor must be taken into account, analysts should recognize its impact, but the BA solution is not particularly dependent on that factor. For instance, the factor "Business Partners" should only be reflected in BA for business strategy.
- 1: The factor is of minor importance and forms only the context of the solution. For instance, "IT infrastructure" is such a factor in the area of BA.

Table 3 constitutes a tool for the management (business management areas defined above are shown in Figures 1 and 2:

- 1: Strategy
- 2: Finance
- 3: Controlling
- 4: Sales

- 5: Purchasing
- 6: Warehousing
- 7: Marketing
- 8: HR
- 9: Assets
- 10: Transport
- 11: Energy
- 12: IT

**Table 3.** Factors of business analytics use in various management fields.

|  | 1 | 2 | 3 | 4 | 5 | 6 | 7 | 8 | 9 | 10 | 11 | 12 |
|---|---|---|---|---|---|---|---|---|---|---|---|---|
| **Business Environment Factors** |  |  |  |  |  |  |  |  |  |  |  |  |
| The Company and its Char. | 3 | 3 | 3 | 3 | 3 | 2 | 3 | 3 | 3 | 2 | 2 | 3 |
| Company Culture | 3 | 3 | 3 | 3 | 2 | 2 | 3 | 3 | 2 | 2 | 2 | 3 |
| Human Resources | 3 | 2 | 3 | 2 | 2 | 2 | 3 | 3 | 2 | 2 | 2 | 3 |
| Legislation | 1 | 3 | 2 | 1 | 2 | 1 | 1 | 2 | 3 | 1 | 2 | 1 |
| Business Partners | 2 | 1 | 1 | 3 | 3 | 2 | 3 | 1 | 2 | 2 | 1 | 3 |
| **Company Environment Factors** |  |  |  |  |  |  |  |  |  |  |  |  |
| Data Sources | 1 | 2 | 2 | 3 | 3 | 2 | 3 | 2 | 3 | 2 | 1 | 3 |
| Application Resources | 2 | 3 | 3 | 3 | 3 | 3 | 2 | 2 | 2 | 2 | 2 | 3 |
| IT Infrastructure | 1 | 1 | 1 | 2 | 2 | 2 | 2 | 1 | 1 | 1 | 2 | 2 |
| State of Business Analytics | 3 | 3 | 3 | 2 | 2 | 2 | 3 | 2 | 2 | 1 | 1 | 2 |
| Applied Approaches | 2 | 3 | 3 | 2 | 2 | 2 | 3 | 2 | 2 | 1 | 1 | 3 |
| **Market Environment Factors** |  |  |  |  |  |  |  |  |  |  |  |  |
| Sourcing | 1 | 2 | 2 | 2 | 2 | 2 | 3 | 2 | 2 | 2 | 1 | 3 |
| Products | 1 | 2 | 2 | 2 | 2 | 2 | 2 | 2 | 2 | 1 | 1 | 2 |
| Cloud Computing | 1 | 2 | 2 | 2 | 1 | 1 | 1 | 1 | 1 | 1 | 1 | 3 |
| Services | 2 | 3 | 3 | 2 | 2 | 2 | 3 | 2 | 2 | 1 | 1 | 3 |
| Availability, Quality and Price Range | 2 | 2 | 2 | 2 | 1 | 1 | 3 | 2 | 1 | 1 | 1 | 3 |

Source: Prepared by the authors.

The defined set of factors, as well as the management fields are based on a general level that should be modified according to reality, current needs, and the business environment of individual companies. On the other hand, estimating the "strength of impact" of a single factor could be effective in the sense that it may help in defining project priorities, potential effects, or risks more accurately.

## 6. Conclusions and Further Research

Business analytics tasks and projects are demanding, not only in terms of the knowledge of BA or BI tools and methods, but also in terms of the need for a very deep comprehension of business, business requests, and the principles of business transformation. Such comprehension must be combined with the knowledge of factors that influence business, management fields, and the use of business analytics in them. This paper attempted to describe one of the many possible approaches to understand these factors. A thorough understanding of these factors could make business analytics solutions more effective and successful.

The next research step will be to focus on the detailed analysis of factors according to different branches of business.

**Author Contributions:** Conceptualization, M.P., J.P. and W.I.; methodology, M.P. and W.I.; validation, M.P., J.P. and W.I.; formal analysis, M.P. and J.P.; resources, M.P.; data curation, M.P., J.P. and W.I.; writing—original draft preparation, M.P.; writing—review and editing, M.P., J.P. and W.I.; supervision, J.P.; project administration, M.P.; funding acquisition, M.P. All authors have read and agreed to the published version of the manuscript.

**Funding:** This research was supported by the Institutional Support for Long-Term and Conceptual Development of Research and Science at the Faculty of Informatics and Statistics, Prague University of Economics and Business (IP400040).

**Acknowledgments:** The support of the employees and management at all the studied companies is gratefully acknowledged.

**Conflicts of Interest:** The authors declare no conflict of interest. The funders had no role in the design of the study; in the collection, analyses, or interpretation of data; in the writing of the manuscript, or in the decision to publish the results.

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
