# Peer review of "Factors Influencing Business Analytics Solutions and Views on Business Problems"

_data, 2071_

Round 1
Reviewer 1 Report
This manuscript provides a really interesting identification of main factors that influence business analytics in Czech businesses. The selected approach and methodology are in accordance with the studied context and the objectives of the research. All identified factors are scored, so that their relevance and importance can be assessed.
The research is scientifically sound and well-presented in the paper.
I just have one suggestion regarding Table 3.
- Is it possible to provide aggregated value for the global factors or another statistics allowing an assessment of their importance for each business area?
A very recent publication in Sustainability, covering the challenge of implementing business analytics in poor contexts, takes up some of your arguments and might be of interest for your study: https://www.mdpi.com/2071-1050/13/9/4882.
Minor comments:
- Line 63: delete “They” at the end of the line
- Line 65-66: There is a repetition
- Table 3: can you put the various business areas in a note below the Table?
Author Response
Dear reviewer,
Thank you for your positive feedback. Below I attach our comments to your feedback.
Suggestion regarding Table 3:
R1: Is it possible to provide aggregated value for the global factors or another statistics allowing an assessment of their importance for each business area?
MP: We have analysed factors of business analytics used in various management fields. From our perspective the impact level should not be aggregated. Each factor is having different impact within the management field.
R1: A very recent publication in Sustainability, covering the challenge of implementing business analytics in poor contexts, takes up some of your arguments and might be of interest for your study: https://www.mdpi.com/2071-1050/13/9/4882.
MP: Thank you for the recommendation. We have added the link to the literature review.
Minor comments:
R1: Line 63: delete “They” at the end of the line
MP: It has been approved according to your comment.
R1: Line 65-66: There is a repetition
MP: It has been approved according to your comment.
R1: Table 3: can you put the various business areas in a note below the Table?
MP: It has been included above the table, just below the related text.
Reviewer 2 Report
I would like to thank the authors for their work and editor for giving me the opportunity to review and interesting well written piece of work.
Here is some specific feedback:
This paper suggests a useful model which documents factors influencing business analytics. This contribution should be clearly outlined in the abstract
Introduction
The introduction is well written and set the scene defining key terms etc. I would like to encourage the authors to provide a stronger problem statement. Why do we need this research? They do mention Gartner statistics which is good, however are there other reasons this work is needed? In addition finish the introduction with a statement outlining the structure of your paper.
Literature review
A neat but short lit review, the discussion around business analytics was good and discusses the purpose and use of BI. It would strengthen the discussion to include background on the business problems, as the title of the paper suggests, although the research questions do not. Perhaps the paper title needs rethinking.
Research methodology
Good introduction, but you don't mention Yin who is well known author for case study research. Additional information is required around how the participants were recruited and selected. Also how many interviews were conducted? How was the data handled? More information about the purpose and origin of table 1 is required. Additional information should be included regarding the theoretical model and analytic approach used to investigate and evaluate the interview data. I believe this the purpose of table 1 and figure 1?
Results and discussion
The discussion provides a model which I assume is the result of analysing the data, this should be made clear. Good discussion of the key aspects of the model, additional literature support through this section would add strength to the discussion and provide validity for the implications section. Table 3 is very good, and a useful summary for anyone wanting to better understand business analytics field and its application. For example from the data the company characteristics are a strong influence across the 12 business content areas identified. Table 2 perhaps needs more explanation regarding its purpose. This is not clear from the content and presentation provided. It would also be could to include any implications for theory.
Overall I really liked this piece of work, and commend the authors efforts. A few areas needs some more editing and clarification. The model developed from analysis of the case study and literature is a useful tool for organisations thinking of using business analytics.
Author Response
Dear reviewer,
Thank you for your positive feedback. Below I attach our comments to your feedback.
R2: This paper suggests a useful model which documents factors influencing business analytics. This contribution should be clearly outlined in the abstract.
MP: It has been approved according to your comment.
Introduction:
R2: The introduction is well written and set the scene defining key terms etc. I would like to encourage the authors to provide a stronger problem statement. Why do we need this research? They do mention Gartner statistics which is good, however are there other reasons this work is needed?
MP: It has been approved according to your comment.
R2: In addition finish the introduction with a statement outlining the structure of your paper.
MP: The introduction has been updated to include a statement outlining the structure of our paper.
Literature review:
R2: A neat but short lit review, the discussion around business analytics was good and discusses the purpose and use of BI. It would strengthen the discussion to include background on the business problems, as the title of the paper suggests, although the research questions do not.
MP: It has been updated according to your comment.
R2: Perhaps the paper title needs rethinking.
MP: Name of the paper remains the same as we updated the introduction.
Research methodology
R2: Good introduction, but you don't mention Yin who is well known author for case study research.
MP: Yes, we do refer to both Myers and Yin.
This application is fully consistent with the definition of a case study as a qualitative research method [31,32].
R2: Additional information is required around how the participants were recruited and selected. Also how many interviews were conducted? How was the data handled?
MP: It has been updated in the text according to your comment.
R2: More information about the purpose and origin of table 1 is required.
MP: More information has been added to the table 1.
R2: Additional information should be included regarding the theoretical model and analytic approach used to investigate and evaluate the interview data. I believe this the purpose of table 1 and figure 1?
MP: It has been updated in the Research Methodology.
Results and discussion:
R2: The discussion provides a model which I assume is the result of analysing the data, this should be made clear.
MP: It has been updated and included factors in the initial part of this section.
R2: Good discussion of the key aspects of the model, additional literature support through this section would add strength to the discussion and provide validity for the implications section.
MP: Additional literature has been added.
R2: Table 2 perhaps needs more explanation regarding its purpose.
MP: It has been updated and linked to the managerial implications section.